# Non-Surgical Interventions for the Prevention of Clinically Relevant Postoperative Pancreatic Fistula—A Narrative Review

**DOI:** 10.3390/cancers15245865

**Published:** 2023-12-15

**Authors:** Nadya Rykina-Tameeva, Jaswinder S. Samra, Sumit Sahni, Anubhav Mittal

**Affiliations:** 1Faculty of Medicine and Health, University of Sydney, Camperdown, NSW 2050, Australia; 2Kolling Institute of Medical Research, University of Sydney, St Leonards, NSW 2065, Australia; 3Upper GI Surgical Unit, Royal North Shore Hospital, St Leonards, NSW 2065, Australia; 4Upper GI Surgical Unit, North Shore Private Hospital, St Leonards, NSW 2065, Australia; 5Australian Pancreatic Centre, St Leonards, NSW 2065, Australia

**Keywords:** clinically relevant postoperative pancreatic fistula, neoadjuvant therapy, somatostatin, somatostatin analogues, antibiotics, analgesia, corticosteroids, nutrition, protease inhibitors, prevention

## Abstract

**Simple Summary:**

Clinically relevant postoperative pancreatic fistula (CR-POPF) continues to be the leading cause of morbidity and mortality after pancreatic surgery. Attempts to prevent the complication have recently expanded to involve non-surgical interventions to supplement surgical solutions and minimise the associated post-pancreatectomy acute pancreatitis. The following narrative review identified ten strategies reported in the literature, including neoadjuvant therapy, somatostatin and its analogues, antibiotics, analgesia, corticosteroids, protease inhibitors, miscellaneous interventions with few reports, combination strategies, nutrition, and fluids. As the reported efficacy of these strategies varied, further higher-powered studies are needed to determine if any of the preventative approaches can be recommended for routine CR-POPF prophylaxis. By reducing CR-POPF, patients may avoid morbid sequelae, experience shorter hospital stays, and ensure timely delivery of adjuvant therapy, overall aiding survival where prognosis, particularly in pancreatic cancer patients, is poor.

**Abstract:**

Clinically relevant postoperative pancreatic fistula (CR-POPF) is the leading cause of morbidity and mortality after pancreatic surgery. Post-pancreatectomy acute pancreatitis (PPAP) has been increasingly understood as a precursor and exacerbator of CR-POPF. No longer believed to be the consequence of surgical technique, the solution to preventing CR-POPF may lie instead in non-surgical, mainly pharmacological interventions. Five databases were searched, identifying eight pharmacological preventative strategies, including neoadjuvant therapy, somatostatin and its analogues, antibiotics, analgesia, corticosteroids, protease inhibitors, miscellaneous interventions with few reports, and combination strategies. Two further non-surgical interventions studied were nutrition and fluids. New potential interventions were also identified from related surgical and experimental contexts. Given the varied efficacy reported for these interventions, numerous opportunities for clarifying this heterogeneity remain. By reducing CR-POPF, patients may avoid morbid sequelae, experience shorter hospital stays, and ensure timely delivery of adjuvant therapy, overall aiding survival where prognosis, particularly in pancreatic cancer patients, is poor.

## 1. Introduction

### 1.1. Clinically Relevant Postoperative Pancreatic Fistula (CR-POPF) 

Pancreatectomy continues to be the mainstay treatment for patients with challenging pancreatic pathology. Pancreatoduodenectomy (PD) is performed for pathology in the head or neck of the pancreas, while distal pancreatectomy (DP) is required for pathology in the body or tail of the pancreas. However, morbidity following these surgeries remains high. One of the major reasons behind this high morbidity rate is clinically relevant postoperative pancreatic fistula (CR-POPF), a complication affecting up to 50% of patients [1]. A pancreatic fistula occurs when the pancreatic ductal epithelium becomes exposed to a new epithelial surface, resulting in the leakage of erosive pancreatic fluid and damage to local tissues [2]. Given that only 20% of pancreatic cancer patients have resectable disease at the time of diagnosis [3], it is imperative that surgical outcomes are optimised to offer these patients and others with a complicated pancreatic disease the best prognosis.

The International Study Group on Pancreatic Surgery (ISGPS) provides the consensus definition for CR-POPF, which encompasses two grades, Grade B and C [4]. Another widely cited definition is that by the American College of Surgeons National Surgical Quality Improvement Program (ACS-NSQIP) [5]. Both definitions are two-fold, requiring first, that drain amylase levels exceed three times the institution’s upper limit of normal for serum amylase and second, a deterioration that alters the patient’s postoperative management. These clinically relevant deteriorations may include haemorrhage and infection (ISGPS Grade B) or, more gravely, sepsis, organ failure, and death (ISGPS Grade C). As such, patients can endure longer, more intensive, and costly hospital stays, readmissions, and delays in adjuvant cancer therapy, worsening prognosis [6,7].

### 1.2. Pathogenesis of CR-POPF

CR-POPF begins with ischaemia, which causes an active pancreas to secrete protease-rich fluid as early as the time of surgery (Figure 1) [8,9]. These proteases, mainly trypsin and chymotrypsin, are activated either locally, by pepsin in gastric secretions, enterokinases in the duodenum, or in acidic environments created by bacteria and fungi [10,11]. Trypsin then continues the activation of proteases, which digest neighbouring tissues, causing erosion, necrosis, and the leakage of pancreatic juice into the abdomen [11], exacerbating the fistula and associated pancreatitis. Local ischaemia also increases capillary permeability, worsening pancreatic juice leakage into neighbouring tissues [12]. This process continues during the initial inflammatory phase of anastomotic healing and wound closure [13]. The enzymatic pathogenesis is accompanied by an independent inflammatory cascade mediated by acinar cells [14]. In CR-POPF patients, this was confirmed by the identification of the inflammatory marker NF-κB at the surgical transection line [15], which is widely implicated in acute pancreatitis [16]. Activation of NF-κB in acinar cells mounts both a local and systemic inflammatory response, independently of trypsinogen [17,18]. Together, these contribute to a symptomatic pancreatic leak, which is ultimately diagnosed as CR-POPF.

### 1.3. Rationale for Exploring Non-Surgical Interventions for the Prevention of CR-POPF

CR-POPF appears primarily a complication of the pancreas itself rather than the surgeon’s technique. This has become increasingly evident as attempts to improve surgical technique and centralise pancreatectomies to high-volume specialty centres have led to minimal improvements in the rates of CR-POPF [7,19]. With pancreatitis and gland activity dictating risk, non-surgical interventions may effectively suppress these triggers and minimise the development of CR-POPF. As such, five databases were searched to investigate this, with relevant abstracts and full texts summarised.

## 2. Literature Search Results

Various non-surgical interventions have been specifically investigated or incidentally found to prevent CR-POPF as a result of their interference with its pathogenesis. These agents can be administered well in advance of surgery, immediately before, during, or after surgery prior to the development of CR-POPF. The literature search identified eight pharmacological interventions, including neoadjuvant therapy, somatostatin and its analogues, antibiotics, analgesia, corticosteroids, protease inhibitors, miscellaneous interventions with few reports, and combination strategies. Nutrition and fluids were identified as two further non-surgical interventions (Appendix A). Meta-analyses investigating somatostatin, pasireotide, octreotide, neoadjuvant therapy, and miscellaneous others were also identified (Table 1, Table 2, Table 3, Table 4 and Table 5). Recommendations for future research have been proposed given these findings (Table 6). 

## 3. Pharmacological Interventions for the Prevention of CR-POPF

### 3.1. Neoadjuvant Therapy 

NAT has not been used with the intention to modulate CR-POPF risk exclusively, but given this advantage, its effects on the complication have been studied. Patients receiving neoadjuvant therapy (NAT), such as chemotherapy (NAC) and radiation (NAR) therapy, have been observed to have more fibrotic parenchyma, in which the risk of CR-POPF is reduced due to lower gland activity [43,44,45]. The subsequent obstructive chronic pancreatitis may increase circulation within the organ by slowing and increasing arterial flow [22], protecting against ischaemia and subsequent enzyme activity. However, it is important to note that patients with locally advanced cancer receiving NAT may by virtue of their disease, have more atrophied and hence fibrotic glands. 

Meta-analyses have found a significantly lower rate of CR-POPF in pancreatectomy patients after NAT (Table 1) [20,21,23,46,47]. Here, chemoradiotherapy (NACR) appeared to provide the greatest benefit [23], particularly in PD patients [47]. Similarly, DP patients receiving NAT did not experience a reduction in CR-POPF [20], while PD patients who did not receive NAT were at an increased risk of CR-POPF [21]. No meta-analyses examining the effect of NAC or NAR alone on CR-POPF have been conducted. 

Intraoperative radiotherapy has also been investigated. Evans et al. reported one case of CR-POPF (2.3%) after PD; however, this may have been confounded by the protective effect of NAC administered to 59% of their patients [48]. Similarly, Bhome et al. investigated intraoperative radiotherapy, finding one case of CR-POPF (5%) in their PD cohort [49]. However, as most patients received neoadjuvant chemotherapy (94.7%), and a control group was not included, the reliability of these findings is not certain. As such, the efficacy of this approach requires further confirmation, particularly its benefits when used in isolation and in combination with NAC. More recently, neoadjuvant ablative stereotactic magnetic resonance-guided adaptive radiation therapy was not found to yield different CR-POPF rates when compared with NACR [50]. Another study adopted this technique after NAC reported a CR-POPF rate of 14% [51]. A randomised controlled trial (RCT) will clarify if this is a reduction to the typical complication rate. 

Future work should seek to compare the effects of NAC, NACR, NAR, and intraoperative radiotherapy on CR-POPF and, within each treatment modality, the regimens that confer the greatest risk reduction after DP and PD. 

### 3.2. Somatostatin Analogues 

Somatostatin analogues inhibit pancreatic exocrine secretions and have hence been hypothesised to be useful in preventing CR-POPF. Somatostatin, octreotide, pasireotide, and lanreotide have been trialled with varying success (Table 5). The selection of somatostatin analogues has been guided by the drugs’ receptor binding profiles, half-life, and cost [52]. Numerous meta-analyses examining the efficacy of somatostatin analogues, including octreotide and pasireotide, have produced conflicting findings (Table 2, Table 3 and Table 4). The incidence of CR-POPF was significantly lower when PD and DP patients were analysed together [31,36,37,53]; however, the certainty of the evidence was classified as low in the most recent meta-analysis [36], and no benefit was found when octreotide was specifically studied [25]. This could represent a change from the earliest meta-analyses, where no difference was reported [26,29,33]. When analysed separately, CR-POPF was found to be lower in DP patients treated with somatostatin [37] but not pasireotide specifically [27]. Further meta-analyses analysing PD patients found benefits in CR-POPF following the prophylactic use of pasireotide [32] but not somatostatins [21,24,32], or octreotide [28,32,39]. Interestingly, the use of postoperative somatostatins was found to be a risk factor for CR-POPF in a meta-analysis of PD patients [21]. 

Octreotide has additionally been hypothesised to reduce fistula formation by increasing pancreatic hardness; however, no statistical difference was observed [54]. A study not included in any meta-analyses by Ecker et al. reported that the omission of octreotide in PD patients was independently associated with decreased CR-POPF [55]. Similarly, Casciani et al. found prophylactic octreotide to be an independent predictor for high (>700 mL) intraoperative blood loss, which itself was an independent predictor for CR-POPF [56]. Moreover, somatostatin analogues may cause hyperglycaemia [57], impair anastomotic healing by decreasing splanchnic blood flow [58], and alter T cell function, impairing postoperative immunity [59]. Considering the risks of somatostatin analogues, conflicting, inconclusive, and heterogenous meta-analyses [36,37], these drugs have not been widely adopted for routine use in pancreatic surgery [55,60,61]. 

### 3.3. Corticosteroids

A meta-analysis on the efficacy of corticosteroids found a significant CR-POPF reduction in patients undergoing pancreatectomy [36] (Table 5).

#### 3.3.1. Dexamethasone

Usually prescribed prophylactically for postoperative nausea and vomiting [62], dexamethasone’s anti-inflammatory effects may reduce CR-POPF and the dissemination of micro-metastases [63,64,65]. Dexamethasone has been trialled to reduce the occurrence of CR-POPF but has not been found to be effective following PD [64,66]. Here, the CR-POPF rates in the control groups were 6% [64] and 6.7% [66], which are lower than the average 17% reported in a recent large meta-analysis [67]. The effects of dexamethasone may have been underestimated by Newhook et al. [66] as patients with pancreatic ductal adenocarcinoma, a pathology known to increase pancreatic fibrosis [68], were enrolled. A total of 75.4% of the patients also received NAT, which decreases acinar content [43], which, together, may have reduced CR-POPF risk. 

Dexamethasone has been found to significantly lower postoperative infections after PD [64], a known risk factor for CR-POPF [69]. Another mechanism through which dexamethasone may protect against CR-POPF is by decreasing the use of opioids [70]. While total morphine equivalents were a significant predictor of CR-POPF after DP [71,72], this has yet to be confirmed in PD patients. Therefore, dexamethasone may curb the development of CR-POPF by reducing inflammation, infection, and opioid use. The promising potential of dexamethasone in reducing CR-POPF requires confirming by RCTs, particularly in high-risk patients. 

#### 3.3.2. Hydrocortisone

The impact of hydrocortisone on CR-POPF has also been studied. A study by Laaninen et al. defined patients at a high-risk of CR-POPF if their intraoperatively determined acinar cell density at the resection border was >40% [73]. In their subsequent work, they found that hydrocortisone decreased the rate of CR-POPF in DP patients [74] but not in PD patients [75]. However, a RCT using the same criteria found that hydrocortisone did decrease CR-POPF in high-risk PD patients [76]. If steroids are indeed effective in minimizing the inflammation that triggers CR-POPF, this may only benefit DP patients, as a failure of the pancreaticoenteric anastomosis may still cause CR-POPF in PD patients.

Hydrocortisone has also been investigated alongside the use of an external pancreatic stent, inner invagination, and Blumgart outer layer [77]. A total of 14 PD patients underwent these interventions with a CR-POPF rate of 7.1%. However, the significance of this could not be determined as there was no control group. As such, these strategies should be validated in a RCT, studying both PD and DP patients and in various combinations to determine the efficacy of each of the strategies and their summative effects.

### 3.4. Analgesia

In an attempt to reduce perioperative opioid use, non-steroidal anti-inflammatory drugs (NSAIDs) have been relied upon as part of effective multimodal analgesia [78]. The most frequently used drugs in pancreatic surgery include indomethacin, ketorolac, lornoxicam, and cyclooxygenase-2 (COX-2) selective drugs. NSAIDs inhibit COX enzymes, which synthesise prostaglandins and thromboxane, causing inflammation which, if excessive, may disturb wound healing [79]. The abstract for a meta-analysis on NSAIDs has recently been published, finding no difference in CR-POPF irrespective of drug selectivity (Table 5) [41]. However, as the full text is not available and to add more clarity to this conclusion, the findings and design of key studies presumed to be included in the meta-analysis are also reviewed.

The study exploring postoperative NSAIDs with the highest number of PD patients found that NSAIDs were associated with significantly greater CR-POPF [80]. In smaller studies considering non-selective NSAIDs, no difference in CR-POPF was reported after PD [81,82,83,84], even in high-risk (>40% acinar cell density) patients [76]. However, in some of these analyses, important confounders such as pancreas texture and duct size were not controlled for, potentially underestimating the harmful effects of NSAIDs through the inclusion of lower-risk patients [81,84]. In a cohort of mixed pancreatectomy patients, ketorolac was significantly associated with CR-POPF [85]. As such, the effect of non-selective NSAIDs on particular pancreatectomy patients requires further clarification in larger, cohort-specific RCTs. Moreover, as non-selective NSAIDs may impair angiogenesis and collagen deposition critical for wound healing, they should be used cautiously, with alternatives preferred if possible [80]. This also applies to COX-2 selective inhibitors associated with CR-POPF, even at low doses [81]. These drugs have added thrombotic effects, which pose a further risk to wound healing [86]. These pathogenic mechanisms appear to support the findings of Behman et al., who reported a significant increase in CR-POPF in PD patients given COX-2 inhibitors but not non-selective inhibitors [81]. As such, a more specific analysis might be necessary to reveal the harms associated with particular NSAIDs. Whilst the risk-benefit profile of NSAIDs following pancreatectomy has yet to be fully established, evidence following bariatric, colorectal and gastroesophageal junction surgery indicates a harmful effect on anastomotic leakage [87,88,89,90,91,92]. Therefore, given the overall assessment of risk based on the current literature, NSAIDs should not be routinely used for analgesia in pancreatectomy patients. 

Other forms of analgesia have also been explored. No difference in CR-POPF was found when PD patients received epidural anaesthesia [93], opiate patient-controlled analgesia (PCA), or both were compared [94]. A comparison of PD patients receiving parecoxib (COX-2 inhibitor) until POD5 or opioids as needed found no difference in CR-POPF [95]. However, potential confounders from this study include the discrepancy in treatment duration, the undisclosed dosage of opioids, and the allowance of opioid analgesia as needed in the former group. This finding requires further confirmation, particularly given the significantly lower serum IL-6 levels in the parecoxib group, which could reflect a helpful modulating effect on PPAP. The parecoxib group also had significantly lower opioid consumption [95]. The impact of opioid use on CR-POPF after PD has not been investigated. Following DP, there was no difference in CR-POPF rates in patients receiving intraoperative and postoperative epidural or intravenous fentanyl PCA [96]. However, other studies have shown that patients who consumed less morphine after DP had less CR-POPF [71,72]. As such, the impact of opiates on CR-POPF requires further clarification, particularly given their adverse effects of respiratory depression, excessive sedation, and intestinal dysmotility [72,83]. While this is being elucidated, multimodal analgesia involving lidocaine, regional nerve blockades, and ketamine has been recommended to reduce opioid consumption [97].

### 3.5. Antibiotics

Infections are increasingly being implicated as a trigger for CR-POPF, with bacteria having recently been found to activate trypsinogen [98,99,100]. With this link evolving, the efficacy of various perioperative regimens has been investigated. PD patients prescribed quinolones were not found to have different rates of CR-POPF [101], but a reduction in CR-POPF was noted when piperacillin–tazobactam was compared with cefoxitin [102] and ampicillin–sulbactam [103]. The additional coverage afforded to *Enterococcus* species by piperacillin–tazobactam may be beneficial as *Enterococcus* spp. were some of the most commonly identified microorganisms in the drain and biliary fluid in pancreatectomy patients with CR-POPF [104]. Specifically, *Enterococcus faecium* was found to be associated with additional postoperative complications [104] and has been cultured significantly more in CR-POPF patients after PD [105,106]. As *Enterococcus faecalis* has been experimentally found to cause intestinal [107] and colonic anastomotic leaks [108], the anti-*Enterococcus* activity of piperacillin–tazobactam may improve healing. *Enterobacter* spp. and *Enterococcus* spp. are cefoxitin-resistant organisms found in bile, with the former identified significantly more in CR-POPF after PD [77], and as such, may also explain the observed superiority of piperacillin–tazobactam [109]. Moreover, as piperacillin–tazobactam was associated with a significant reduction in surgical site infections [102], curbing infective complications may, in turn, avoid the formation and exacerbation of CR-POPF for which it is a risk factor [110]. However, no effect was found in PD patients with positive preoperative bile cultures [111], undergoing preoperative biliary drainage [112], or high-risk patients receiving piperacillin–tazobactam and gentamicin prophylaxis for 2 to 5 days postoperatively [113]. These studies may identify a cohort of patients who are more vulnerable to CR-POPF and in whom additional antibiotic strategies are required.

The efficacy of different generations of cephalosporins has also been explored. Kondo et al. compared prophylactic cefmetazole (second generation) with piperacillin, finding no difference in the rate of infected pancreatic fistula [114]. However, a minority of patients in each group received other antibiotics, and as this form of fistula is only a subset of possible CR-POPF cases, the total effect of these antibiotics on CR-POPF cannot be concluded. After PD, two studies reported no difference when first- to third-generation cephalosporins were trialled [115,116]; however, one study found greater CR-POPF in patients receiving first-generation cephalosporins but not after DP [117]. The safety of later-generation cephalosporins appears to be supported by further studies where replacing first- and second-generation cephalosporins with a third-generation cephalosporin alongside metronidazole reduced CR-POPF after PD [118,119,120]. 

Studies have also administered antibiotics for varying durations. When comparing perioperative and prolonged prophylaxis, no difference in CR-POPF has been found in PD patients nor in patients who have undergone preoperative biliary drainage [93,121,122,123,124,125]. Droogh et al. compared cefazolin (first generation) and metronidazole with cefuroxime (second generation) and metronidazole. In the initial regimen, patients were treated every 4 h intraoperatively, whereas for the alternate regimen, patients were treated three times a day until POD5 [124]. No difference in CR-POPF was reported. Another study that administered cefmetazole (second generation) intraoperatively until POD1 or POD2 found no difference in CR-POPF [125]. This may be explained by the possibility that extending the use of first-generation cephalosporins may increase their safety, while no added benefit is seen with longer use of second-generation cephalosporins. 

Fourth-generation cephalosporins have also been explored in patients who have undergone biliary drainage. Cefozopran (fourth generation) given for 1 day had significantly less CR-POPF when compared to the group treated until POD4 [126]. This advantage may, however, have been exaggerated by the significantly higher BMI in the latter group, which is an established risk factor for CR-POPF [47]. In a similar study, cefazolin (first generation) was given to patients without preoperative biliary drainage, while cefozopran was given to internal biliary drainage patients, and targeted antibiotics were given to external drainage patients. Here, infection rates in the drainage patients were comparable, with no difference in CR-POPF found for any treatment [127]. An important caveat for these studies is that preoperative biliary drainage has been identified as protective against CR-POPF after PD [128], potentially overestimating the reported benefits of the antibiotics.

Intraabdominal administration of antibiotics has also been trialled. The intraoperative irrigation of anastomoses with polymyxin B during PD made no difference in CR-POPF when compared with saline [129]. Imipenem and vancomycin have also been studied, with one group administering these antibiotics postoperatively via a drain at the pancreatojejunostomy anastomosis. Here, there was no difference in CR-POPF; however, only 10 patients were in the study and control groups, and so may have been underpowered [130]. This raises the potential to explore the efficacy and practicality of various routes of administration for antibiotics, particularly as intraabdominal administration appears to have the additional benefit of allowing for the delivery of higher concentrations of antibiotics.

Therefore, the effect of perioperative and prolonged prophylaxis with different cephalosporins and the effect of metronidazole on CR-POPF risk remains to be clarified in PD cohorts who have and have not undergone preoperative biliary drainage. Similarly, DP cohorts must be examined as most studies thus far have focused only on PD patients. Given the distinct microbial profiles seen in PD and DP patients [131], a targeted antimicrobial prophylaxis approach may be warranted based on the findings of these future studies. Further questions to answer would be the ideal duration of antibiotic prophylaxis and the efficacy of giving antibiotics in response to high drain amylase, as significant regional discrepancies have been reported for these variables [132].

### 3.6. Protease Inhibitors

Currently, protease inhibitors appear insufficient alone in managing the multiple inflammatory mechanisms that contribute to acute pancreatitis and may be incapable of penetrating the pancreas sufficiently during inflammation [133]. As such, they have been investigated as part of singular and combination strategies for reducing CR-POPF. Ulinastatin, a trypsin inhibitor, has been found to significantly reduce CR-POPF after PD in a meta-analysis combining two studies (Table 5) [30]. Its mechanism of inactivating pancreatic juice may also explain the noted reduction in PPAP after PD [134]. As such, further investigations of these agents may be warranted, given the evolving understanding of the interplay between pancreatitis and CR-POPF. No Grade C POPF was reported when gabexate mesilate was used to irrigate the pancreaticojejunostomy anastomosis after PD. However, only 27 patients were included, and no control was used [135], with no studies confirming this finding since. Gabexate mesilate has been trialled as part of triple therapy, also including octreotide and carbapenem antibiotics. DP patients with high amylase output underwent drain removal on POD1 or after POD5 and received this triple therapy, finding no exacerbation of CR-POPF in the POD1 group [136,137]. However, early drain removal (before POD3) after DP has been shown to be associated with less CR-POPF [138]. The same group repeated this in PD patients with high drain amylase on POD1, with most patients having their drains removed from POD5 onwards. Here, there was a significant reduction in CR-POPF compared to patients who did not receive the triple therapy, irrespective of their drain amylase level [137]. The standardisation of drain protocols adds confidence to this finding and highlights the distinct character of CR-POPF after each pancreatectomy type. Another study in DP patients investigated the effects of octreotide while also providing gabexate mesilate to all patients. No difference in CR-POPF was found; however, it is unclear how this dual regimen may have confounded the findings [139]. Overall, the efficacy of gabexate mesilate alone in reducing CR-POPF remains unclear.

### 3.7. Emerging Pharmacological Interventions

#### 3.7.1. Interventions with Few Reports 

Preoperative botulinum toxin has been trialled to relax the sphincter of Oddi after DP, significantly reducing CR-POPF [140]. However, this was not confirmed in a validation study [141]. This discrepancy may have been owed to the smaller sample size and later delivery of botulinum toxin, where the median was one day before surgery compared to six days in the original report. Hence, clarifying the ideal timing of botulinum toxin for the prevention of CR-POPF and an assessment of its impact on PD patients is required. Secretin, a hormone that promotes pancreatic bicarbonate release, was not found to affect CR-POPF after pancreatectomy. Hypertension medications after PD were not found to influence CR-POPF [142].

#### 3.7.2. Interventions to Minimise PPAP and by Extension, CR-POPF

It can be hypothesised that preventing PPAP may help reduce CR-POPF. Lornoxicam was found to significantly reduce PPAP after pancreatectomy [143], highlighting an opportunity to better characterise the relationship between CR-PPAP and CR-POPF. A meta-analysis of pharmacological interventions for acute pancreatitis found very low- or low-quality evidence for the following reducing short-term mortality: lexipafant, octreotide, somatostatin and omeprazole, somatostatin, and ulinastatin [144]. However, no consistent findings for these drugs or their effects were differentiated for patients with mild or severe acute pancreatitis. While some of these drugs have been trialled for CR-POPF, the merit of those that have not cannot be currently recommended on the basis that they might reduce complication rates by targeting PPAP.

The postoperative systemic inflammatory response begins with hyperinflammation [145] and is resolved through immunosuppression. Whilst this sequence has largely been used to describe the pathogenesis of SIRS and infectious complications [145], it might be plausible to apply this theory to the local mechanisms that contribute to PPAP. Numerous interventions may serve to mitigate damage caused by an excessive initial inflammatory response or failure of the secondary immunosuppressive stage. Hyperbaric oxygen therapy has been trialled after PD, finding no difference in the rate of POPF. The promising reduction in pro-inflammatory cytokines IL-6 and IL-10 highlights the potential for this treatment to modulate CR-PPAP and CR-POPF and, as such, should be re-investigated for its effects on these complications [146]. It has been hypothesised that propofol, local anaesthetics, NSAIDs, ketamine, and midazolam preserve immune function [97]. This finding has primarily been related to its subsequent protection against cancer recurrence, with propofol-based intravenous anaesthesia for pancreatectomy having been associated with better survival when compared with volatile-based anaesthetic desflurane [147]. To this principle, these anaesthetic drugs may also be helpful in initiating healing. Intravenous lidocaine and regional anaesthesia have been associated with reduced proinflammatory mediators immediately after pancreatectomy and 24 h later [148]. However, the significance of this for PPAP, particularly given the need for an initial inflammatory response, is not known. By better characterising this, their use can be tailored in cases where excessive inflammation is anticipated. Other agents yet to be investigated but serve the function of decreasing inflammation include antioxidants such as anaesthetics with strong antioxidant effects, namely propofol [149], platelet-activating factor inhibitors, and TNF-α antibodies [150]. Future studies may seek to establish which of these agents support the body’s healing response to surgery by including CR-PPAP as a primary outcome, which may, in turn, clarify the local and systemic inflammatory pathogenesis behind CR-POPF.

### 3.8. Potential Leads from Different Surgical and Experimental Contexts 

Interventions that have been reported to reduce anastomotic leakage following other gastrointestinal surgeries may provide new options for CR-POPF. Phenytoin has been trialled for fistulae occurring in a mixed cohort of gastrointestinal surgeries where it was found to significantly reduce output [151]. By inhibiting collagenase [152], phenytoin can further facilitate healing through fibrosis of the fistula. 

Promising findings from animal studies may also provide new solutions to consider. Beagle dogs treated with fibroblast growth factor in gelatin hydrogel microspheres trialled were found to be protected from pancreaticojejunostomy failure. The noted increase in collagen, fibroblast proliferation, microvessel density, and inflammatory cells was hypothesised to accelerate the closure of the anastomosis after PD [153,154]. Pigs injected with N-butyl-2-cyanoacrylate and lipiodol were reported to have localised increased pancreatic fibrosis and hardness when compared to saline. As this effect was preserved into the first postoperative week [155], it could be effective at curbing the early development of CR-POPF in at-risk patients, namely those with soft pancreases. A similar rationale applies to the injection of the TGF-beta 1 receptor inhibitor with penicillin G. When tested in mice, pancreas texture was found to become significantly harder within 3 days, with this effect reversed after 7 days [156]. Assessments of the long-term impact of these injections on exocrine and endocrine function are required to better characterise their safety profile. 

N-acetyl cysteine works as an antioxidant by neutralising reactive oxygen metabolites from inflamed cells, inhibits NF-κB activation, and reduces oedema, necrosis, and haemorrhage [157]. N-acetyl cysteine has been shown to reduce ischemia–reperfusion injury in transplantation rat models [158] but has not yet been shown to reduce cellular damage associated with acute pancreatitis [159]. As such, it would be valuable to establish its efficacy in CR-POPF animal models.

## 4. Non-Pharmacological Interventions for the Prevention of CR-POPF

### 4.1. Nutrition

Patients with malnutrition, sarcopenic obesity, and lower prognostic nutritional indexes are at a higher risk of CR-POPF [160,161]. Prehabilitation strategies involving nutritional support, optimisation of diabetes and exocrine insufficiency, and respiratory training have been trialled [162], with a meta-analysis finding that prehabilitation may improve CR-POPF (Table 5) [40]. 

Supplements have also been studied. A study of oral preoperative interventions found no difference in CR-POPF when patients did or did not receive synbiotics (Bifidobacterium breve, Lactobacillus casei, and galacto-oligosaccharides) for prevention of bacterial translocation to mesenteric lymph nodes [163]. Two studies examined the effects of enteral immunonutrition in which feeds were supplemented with additional nutrients. Patients receiving enteral feeds enriched with arginine, ω-3 fatty acids, and dietary nucleotides after PD did not have different rates of CR-POPF compared to patients receiving standard enteral nutrition [164]. A similar non-significant result was reported when patients were given these enriched feeds preoperatively and compared with patients who had normal oral intake [165]. However, as CR-POPF was not a primary outcome in either of these studies, they may have been underpowered as there were no more than 10 CR-POPF patients in the investigation groups. As such, studies of sufficient power are required to clarify the role of supplementation in modulating CR-POPF.

### 4.2. Fluids 

The effect of fluid optimisation and blood transfusions on CR-POPF have been investigated. Postoperative blood transfusions have been found to predict prolonged CR-POPF [166], likely due to this reflecting blood loss and subsequent ischaemia. However, perioperative transfusions in a mixed cohort receiving NAT for pancreatic ductal adenocarcinoma found no difference [167], potentially due to these two factors being protective for CR-POPF. 

When intrathecal morphine was given for major hepato–pancreato–biliary surgery, patients experienced less postoperative hypotension and decreased fluid demand when compared with continuous thoracic epidural analgesia [168]. It would be helpful to clarify this finding in specific pancreatectomy cohorts. While increased opioid consumption was a risk factor for CR-POPF after DP [71], its effects in PD patients should be studied, particularly given the potentially protective element of improved fluid balance opioids may offer. Indeed, PD patients who do not receive adequate intraoperative fluids have a higher risk of CR-POPF [169,170]. Conversely, when three regimens of varying restrictiveness for intraoperative intravenous fluid were trialled in PD, no difference in CR-POPF was noted [171].

More locally, irrigation of the peripancreatic area to dilute pancreatic juice to remove necrotic tissue and bacteria has significantly reduced CR-POPF in a recent meta-analysis [42]. 

## 5. Limitations of the Narrative Review

The interventions included in the search strategy did not encompass all those ultimately described in the review. This was overcome by citation chaining to maximise the number of relevant additional studies. Secondly, the findings of this narrative review were often limited by underpowered studies where CR-POPF was not the primary outcome. This will hopefully be overcome by the suggested areas for future research. Furthermore, many studies published after consensus definitions had been established for CR-POPF included biochemical leaks in their analysis, limiting the clinical relevance of their findings. This, unfortunately, prevented deeper analysis and confirmation of the efficacy of the interventions, which would have bolstered the strength of recommendations made in this review. Finally, as many studies included patients with a variety of diagnoses, specific analyses on the efficacy of these interventions on pancreatic cancer patients were not possible. As such, future studies should endeavour to establish the applicability of non-surgical interventions for the prevention of CR-POPF in patients with specific diagnoses. 

## 6. Conclusions

The evolving link between ischaemia, inflammation, CR-POPF, and PPAP has allowed for a transition in our approach to minimising CR-POPF from surgical refinement to non-surgical interventions. As such, these interventions may help reduce the rate of CR-POPF. The present study has provided a comprehensive review of the successes, failures, and remaining questions that pertain to the non-surgical prevention of CR-POPF, allowing numerous opportunities for further research to be identified (Table 6). By further clarifying the value of these interventions for specific patient populations, such as according to their surgical and risk profiles, outcomes for patients with cancer and other challenging pancreatic pathologies can be ultimately improved. 

## Figures and Tables

**Figure 1 cancers-15-05865-f001:**
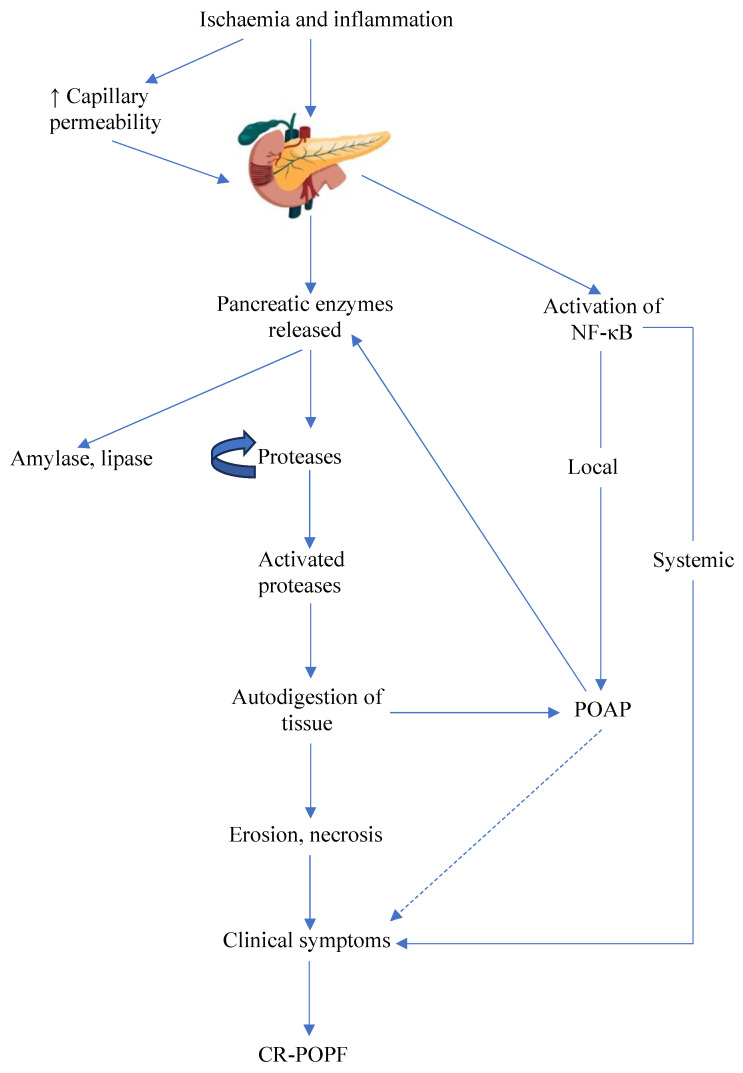
Pathogenesis of CR-POPF. CR-POPF is initiated by ischaemia and inflammation, triggering further inflammation directly through the activation of NF-κB and indirectly through the digestive action of pancreatic proteases. NF-κB contributes to local and systemic inflammation, culminating in the manifestation of clinical symptoms and, eventually, CR-POPF. The release of pancreatic enzymes, and the activation specifically of proteases locally, in the duodenum, by pathogens and through auto-activation (indicated by the self-reflexive arrow), results in the erosion of neighbouring tissues, spreading and worsening the impact of the pancreatic leak. If left unmanaged, clinical symptoms may develop, resulting in CR-POPF. The PPAP that both these concurrent processes contribute to may independently trigger CR-POPF. CR-POPF: clinically relevant postoperative pancreatic fistula; PPAP: post-pancreatectomy acute pancreatitis.

**Table 1 cancers-15-05865-t001:** Summary of meta-analyses reporting on neoadjuvant therapy.

Study	Surgical Cohort	Patients (n)	Pharmacological Intervention	Efficacy of Intervention on CR-POPF
Kamarajah 2020 [20]	Mixed	19,893	NAT	After PD: NAT vs. control OR 0.55, 95%CI 0.46–0.71, *p* < 0.001 After DP: NAT vs. control OR 0.79, 95%CI 0.29–2.20, *p* = 0.7
Kamarajah 2021 [21]	PD	52,774	NAT	CR-POPF risk factors: no NAT OR 0.25, 95%CI 0.14–0.44, *p* < 0.001
van Dongen 2021 [22]	Mixed	25,389	NAT	NAT vs. UFS: lower rate of CR-POPF pooled OR 0.47, 95%CI 0.38–0.58 The effect was mostly due to NACR OR 0.46, 95%CI 0.29–0.73, not NAC alone OR 0.83, 95%CI 0.59–1.16
Vasavada 2022 [23]	Mixed	17,021	NAT	NAT decreased CR-POPF RR 0.66, 95%CI 0.55–0.81, *p* < 0.0001
Zhang 2022 [20]	PD	24,740	NACR	NACR vs. UFS: OR 0.68, 95%CI 0.5–0.81, *p* < 0.001

PD: pancreatoduodenectomy; CR-POPF: clinically relevant postoperative pancreatic fistula; NAT: neoadjuvant therapy; NAC: neoadjuvant chemotherapy; OR: odds ratio; RR: risk ratio; CI: confidence interval; NACR: neoadjuvant chemoradiotherapy; UFS: upfront surgery.

**Table 2 cancers-15-05865-t002:** Summary of meta-analyses reporting on somatostatin.

Study	Surgical Cohort	Patients (n)	Pharmacological Intervention	Efficacy of Intervention on CR-POPF
Adiamah 2019 [24]	PD	1615	Somatostatin	Somatostatin analogue vs. control: OR 0.48, 95%CI 0.22–1.06, *p* = 0.07
Cecire 2022 [25]	Mixed	724	Octreotide	Octreotide vs. control: *p* = ns
Connor 2005 [26]	Mixed	1918	Somatostatin	Somatostatin vs. control: OR 0·80, 95%CI 0·44–1·45, *p* = 0.459
Dalton 2020 [27]	Mixed	1571	Pasireotide	Pasireotide vs. control: Pancreatectomies: *p* = 0.29 DP: OR 0.70, 95%CI 0.30–1.63, *p* = 0.41 PD: OR 0.60, 95%CI 0.42–0.86, *p* = 0.006 (decrease)
Garg 2018 [28]	PD	959	Octreotide	Perioperative octreotide vs. control: OR 0.76, 95%CI 0.35–1.65, *p* = 0.49
Gurusamy 2013 [29]	Mixed	2143	Somatostatin	Somatostatin analogues vs. control: RR 0.69, 95%CI 0.34–1.41 (*p* = ns)
Halle-Smith 2022 [30]	PD	7512	Somatostatin	CR-POPF not reported on
Han 2017 [31]	Mixed	1703	Somatostatin	Somatostatin vs. control: RR 0.60, 95%CI 0.36–0.98, *p* = 0.02
Jin 2015 [32]	PD	1352	Somatostatin	Prophylactic somatostatins vs. control: RR 0.70, 95%CI 0.47–1.050, *p* = 0.08 Somatostatin vs. control: Z test for pooled effect size 1.87, *p* = 0.06
Kamarajah 2021 [21]	PD	52,774	Somatostatin	CR-POPF risk factors: postoperative somatostatin analogues OR 3.24, 95%CI 1.84–5.70, *p* < 0.001
Koti 2010 [33]	Mixed	2143	Somatostatin	Somatostatin analogues vs. control: RR 0.69, 95%CI 0.34–1.41, *p* = 0.31
Li 2020 [34]	Mixed	2221	Somatostatin	Somatostatin vs. control: OR 0.53, 95%CI 0.34–0.83, *p* < 0.01 (decrease)
Liu 2020 [35]	Mixed	919	Pasireotide	Pasireotide vs. control: OR 0.78, 95%CI 0.49–1.24; *p* = 0.29
Rompen 2023 [36]	Mixed	3742	Somatostatin	Somatostatin vs. control: *p* < 0.05 (decrease)
Schorn 2020 [37]	Mixed	1940	Somatostatin	Somatostatin analogues vs. control: PD: RR 0.69, *p* = 0.30 Mixed cohort: RR 0.47, *p* = 0.02 (decrease) DP: RR 0.41, *p* = 0.03 (decrease)
Wang 2017 [38]	Mixed	1902	Octreotide	Octreotide vs. control: RR 0.91, 95%CI 0.55–1.4, *p* = 0.71
Zheng 2019 [39]	Mixed	2006	Octreotide	Octreotide vs. control: RR 1.01, 95%CI 0.68–1.50, *p* = 0.95

PD: pancreatoduodenectomy; CR-POPF: clinically relevant postoperative pancreatic fistula; OR: odds ratio; RR: risk ratio; CI: confidence interval; ns: not significant; (decrease) indicates a decrease in CR-POPF.

**Table 3 cancers-15-05865-t003:** Summary of meta-analyses reporting on octreotide.

Study	Surgical Cohort	Patients (n)	Efficacy of Intervention on CR-POPF
Cecire 2022 [25]	Mixed	724	Octreotide vs. control: *p* = ns
Garg 2018 [28]	PD	959	Perioperative octreotide vs. control: OR 0.76, 95%CI 0.35–1.65, *p* = 0.49
Jin 2015 [32]	PD	1352	Octreotide vs. control: Z test for pooled effect size 0.54, *p* = 0.59
Wang 2017 [38]	Mixed	1902	Octreotide vs. control: RR 0.91, 95%CI 0.55–1.49, *p* = 0.71

PD: pancreatoduodenectomy; CR-POPF: clinically relevant postoperative pancreatic fistula; OR: odds ratio; RR: risk ratio; CI: confidence interval; ns: not significant.

**Table 4 cancers-15-05865-t004:** Summary of meta-analyses reporting on pasireotide.

Study	Surgical Cohort	Patients (n)	Efficacy of Intervention on CR-POPF
Dalton 2020 [27]	Mixed	1571	Pasireotide vs. control: Pancreatectomies: OR 0.84, 95%CI 0.60–1.16, *p* = 0.29 DP: OR 0.70, 95%CI 0.30–1.63, *p* = 0.41 PD: OR 0.60, 95%CI 0.42–0.86, *p* = 0.006 (decrease)
Jin 2015 [32]	PD	1352	Pasireotide vs. control: Z test for pooled effect size 2.19, *p* = 0.03 (decrease)
Liu 2020 [35]	Mixed	919	Pasireotide vs. control: OR 0.78, 95%CI 0.49–1.24; *p* = 0.29

PD: pancreatoduodenectomy; DP: distal pancreatectomy; CR-POPF: clinically relevant postoperative pancreatic fistula; OR: odds ratio; CI: confidence interval; (decrease) indicates a decrease in CR-POPF.

**Table 5 cancers-15-05865-t005:** Summary of meta-analyses on miscellaneous interventions.

Study	Surgical Cohort	Patients (n)	Pharmacological Intervention	Efficacy of Intervention on CR-POPF
Bundred 2020 [40]	Mixed	193	Prehabilitation	Early evidence demonstrates that prehabilitation programmes may improve CR-POPF
Halle-Smith 2022 [30]	PD	7512	Ulinastatin, antibiotic irrigation, total parental nutrition, analgesia, octreotide, somatostatin	Ulinastatin was associated with reduced rates of CR-POPF OR 0.24, 95%CI 0.06–0.93 Meta-analysis did not report any difference for the other interventions
Rompen 2023 [36]	Mixed	3742	Glucocorticoids	Glucocorticoids vs. control: *p* < 0.05 (decrease)
Thomas 2023 [41]	PD	2851	NSAIDs	Non-selective NSAIDs vs. control: OR 1.04, 95%CI 0.66–1.64, *p* = 0.87 COX-2 Inhibitors vs. control: OR 1.52, 95%CI 0.58–3.99, *p* = 0.40
Pergolini 2023 [42]	Mixed	196	Peripancreatic continuous irrigation	Reduced rate of CR-POPF (*p* < 0.05)

PD: pancreatoduodenectomy; DP: distal pancreatectomy; CR-POPF: clinically relevant postoperative pancreatic fistula; OR: odds ratio; CI: confidence interval; NAC: neoadjuvant chemotherapy; NSAIDs: non-steroidal anti-inflammatory drugs; (decrease) indicates a decrease in CR-POPF.

**Table 6 cancers-15-05865-t006:** Recommended future studies to examine the effect of pharmacological interventions on CR-POPF. Interventions should examine both pancreatoduodenectomy and distal pancreatectomy patients separately unless otherwise specified.

Recommendations for future research
(a) **Interventions requiring meta-analyses**
NAC
NAR—including neoadjuvant ablative stereotactic magnetic resonance-guided adaptive radiation therapy and intraoperative techniques
Antibiotics ● Piperacillin-tazobactam in PD patients
Early enteral vs. total parenteral nutrition
(b) **Interventions requiring further RCTs**
Specific regimens for NAC, NACR, NAR, and intraoperative radiotherapy
Dexamethasone
Hydrocortisone including its use in the following combination strategies: Lemke 2021: 1. External pancreatic stent; 2. Perioperative hydrocortisone; 3. Inner invaginating layer; 4. Blumgart outer layer. Kriger 2020: 1. Perioperative hydrocortisone; 2. Somatostatin.
Analgesia types ● NSAIDs: Non-selective vs. COX-2 selective; ● NSAIDs vs. opioids. Analgesia routes ● Epidural vs. intravenous vs. patient-controlled analgesia.
Antibiotic types ● Piperacillin-tazobactam in DP patients; ● First versus third-generation cephalosporins; ● Cephalosporins plus metronidazole. Antibiotic prescribing ● Optimal treatment duration; ● Intraabdominal administration vs. intravenous; ● Tailoring approach to risk profile (e.g., amylase level).
Gabexate mesilate, including its use in the following combination strategy: Adachi 2015 and 2019: 1. Gabexate mesilate; 2. Octreotide; 3. Carbapenem antibiotics.
Nutritional support: ● Preoperative nutrition optimisation in high-risk patients; ● Immunonutrition regimens administered enterally or parenterally.
Botulinum toxin: ● In PD patients; ● Ideal timing in DP and PD patients.
Secretin
(c) **Interventions to explore given their promise in the context of PPAP**
Lornoxicam
Hyperbaric oxygen therapy
Anaesthetic agents capable of facilitating the hyperinflammatory phase of healing ● Propofol; ● Local anaesthetics; ● Ketamine.
Anaesthetic agents that may assist in the immunosuppressive phase of healing ● Intravenous lidocaine; ● Regional anaesthesia.
Anaesthetic agents with antioxidant effects ● Propofol; ● Platelet-activating inhibitors; ● TNF-α antibodies.
(d) **Interventions to explore given their promise in other surgical contexts**
Phenytoin
(e) **Interventions from experimental contexts that may be suitable for trial in humans**
Fibroblast growth factor injections
N-butyl-2-cyanoacrylate and lipiodol injections
TGF-beta 1 receptor inhibitor and penicillin G injections
N-acetyl cysteine

RCT: randomised controlled trial; NAC: neoadjuvant chemotherapy; NACR: neoadjuvant chemoradiotherapy; PPAP: post-pancreatectomy acute pancreatitis.

## Data Availability

No new data were created or analysed in this study. Data sharing is not applicable to this article. All relevant data to this work are contained within the article.

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
