# Peer review of "Non-Surgical Interventions for the Prevention of Clinically Relevant Postoperative Pancreatic Fistula—A Narrative Review"

_cancers, 2023, doi:10.3390/cancers15245865_

Round 1

Reviewer 1 Report

Comments and Suggestions for Authors

A nicely written comprehensive review regarding pancreatic leak complications

Only criticism I have is that the topic discussed is not exclusive to pancreatic cancer. While the special issues is regarding surgical care, it would be important for authors to point out the differences between surgery for pancreatic cancer vs. non-cancer in all sections of the review.

Another food for thought is whether there are differences from one surgeon to another and institution to institution regarding severerity/prevalence of fistula. Is it preventable?

Author Response

Thank you for your feedback.

The authors would like to note that the current literature on CR-POPF reports on cohorts comprised of majority, though not exclusively, pancreatic cancer patients. This is understandable given the dismal prognosis of pancreatic cancer and the subsequent priority of maximising postsurgical outcomes to achieve to the fullest, this prognosis. As such, it unfortunately would not be possible given the available literature, to delineate outcomes of the identified interventions for patients with and without pancreatic cancer. We agree this would be a valuable comparison and hope future studies will facilitate this. Accordingly, we have included this as an additional limitation of our study in lines 608-612, stating: “Finally, as many studies included patients with a variety of diagnoses, specific analyses on the efficacy of these interventions on pancreatic cancer patients was not possible. As such, future studies should endeavor to establish the applicability of non-surgical interventions for the prevention of CR-POPF in patients with specific diagnoses.”

Thank you also for your pondering on whether CR-POPF is preventable as our review has sought to answer this question. The potential for differences in CR-POPF rates between surgeons and institutions has been previously identified and since minimised through centralising pancreatectomies to large volume centres. At these centres, surgeons maintain a high standard of operative technique by ensuring they perform a certain number of pancreatectomies annually. Despite attempts to optimise surgical technique, part of which has involved ensuring adequate case volume, CR-POPF remains the leading cause of post-pancreatectomy morbidity and mortality. We have now explicitly noted this important element of inter-surgeon and institution variability in lines 96-98: “This has become increasingly evident as attempts to improve surgical technique and centralise pancreatectomies to high-volume specialty centres has led to minimal improvements in the rates of CR-POPF [19, 20].” This observation motivated our review and is explained in our rationale under Section 1.3. By shifting our focus to non-surgical strategies for CR-POPF, particularly given its complex biochemical origins (section 1.2), we hope to raise the profile of these lesser known, but promising approaches which may better prevent CR-POPF.

Reviewer 2 Report

Comments and Suggestions for Authors

The Authors conducted  a narrative review of non-surgical interventions for the prevention of clinically relevant pancreatic fistula (CR-POPF). The literature search provided a series of non-surgical procedures specific (somatostatin analogues, corticosteroids, analgesic, antibiotics, protease inhibitors) or aspecific (neoadjuvant therapy) and non pharmachological  products with contrasting results. Ultimately, the etiology of CR-POPF is multifactorial and, it is very difficult for the readers to look about several products with proven efficacy. Moreover, the incidence of CR-POPF  appears to be influenced by pancreatic features, surgical technique and interventions rather than non-surgical procedures.

Author Response

Thank you for your feedback.

We agree the development of CR-POPF is multifactorial, with surgical and pancreatic factors influencing pancreatectomy outcomes. However, despite the predominant focus on surgical optimisation to prevent CR-POPF in recent decades, CR-POPF remains the leading cause of post-pancreatectomy morbidity and mortality. This observation motivated our review and is explained in our rationale under Section 1.3. Reporting on non-surgical strategies for the prevention of CR-POPF allowed us to incorporate the current understanding of its pathogenesis, which is largely biochemical and pancreas dependent (Section 1.2). To help readers understand the efficacy of each intervention, we identified potential confounders and study limitations. Through this, we aimed to describe which non-surgical interventions are effective and in which contexts, to inform future studies and ultimately, best practice decisions.

Reviewer 3 Report

Comments and Suggestions for Authors

This is a narrative review of non-surgical management of POPF. 

1. The authors discussed neoadjuvant therapy but I don't think NAT is performed to prevent CR-POPF. In my personal opinion, this part should be deleted.

2. Similar to above, there is a potential bias that NAT is perofmed in more advanced pancreatic cancer. As seen in no change in CR-POPF after distal pancreatectomy, reduced CR-POPF after NAT was the pancreatic body/tail atrophy in pancreatic head cancer. In this point, NAT patients might have advanced disease and more atrophy in the body and tail. Thus, the atrophy itself, rather than NAT, might be associated with reduced CR-POPF.

3. Does "mixed" in tables mean PD and DP? or other than pancreatectomies?

4.  Tables. P value alone does not represent study data. Please add the numbers or risk/odds ratio.

5. Similar to above, why not conduct a meta-analysis, not a narrative review?

Author Response

Thank you for your feedback.

  1. We agree and have similarly stated this in Section 3.1, lines 136-137: “NAT has not been used with the intention to exclusively modulate CR-POPF risk, but given this advantage, its effects on the complication have been studied.”
  2. Thank you for highlighting this potential caveat. We have included this point in lines 142-144 of Section 3.1: “However, it is important to note that patients with locally advanced cancer receiving NAT, may by virtue of their disease, have more atrophied and hence fibrotic glands.”.
  3. “Mixed” means studies including both PD and DP patients.
  4. We chose to report p-values as this measure was most frequently provided across the studies. To ensure adequate context for these values, we provided the number of patients included in each study. We have sought to include further measures of efficacy where possible (e.g. OR and RR) and have updated our tables accordingly.
  5. Where meta-analyses had been performed for particular non-surgical interventions, we summarised their findings in Tables 1-5. This allowed us to compile and compare their findings. For the remaining interventions, there was an insufficient number of studies for new meta-analyses to be conducted. As such, writing a narrative review allowed us to identify areas for future research after which new or further meta-analyses may be possible.

Round 2

Reviewer 2 Report

Comments and Suggestions for Authors

The paper has been revised. I have no further comments.

Reviewer 3 Report

Comments and Suggestions for Authors

The paper has been revised according to the concerns. I have no further comments.